# A Novel Approach for the Detection of Standing Tree Stems from Plot-Level Terrestrial Laser Scanning Data

**Wuming Zhang [1], Peng Wan [1,*], Tiejun Wang [2], Shangshu Cai [1], Yiming Chen [1], Xiuliang Jin [3] and Guangjian Yan [1]**

[1] State Key Laboratory of Remote Sensing Science, Jointly Sponsored by Beijing Normal University and Institute of Remote Sensing and Digital Earth of Chinese Academy of Sciences; Beijing Engineering Research Centre for Global Land Remote Sensing Products, Institute of Remote Sensing Science and Engineering, Faculty of Geographical Science, Beijing Normal University, Beijing 100875, China; wumingz@bnu.edu.cn (W.Z.); sscai1992@163.com (S.C.); cym_bnu@mail.bnu.edu.cn (Y.C.); gjyan@bnu.edu.cn (G.Y.)

[2] Faculty of Geo-Information Science and Earth Observation (ITC), University of Twente, P.O. Box 217, 7500 AE Enschede, The Netherlands; t.wang@utwente.nl

[3] INRA-EMMAH, UMT-CAPTE, 84914 Avignon, France; jinxiuxiuliang@163.com

[*] Correspondence: wanpeng@mail.bnu.edu.cn; Tel.: +86-189-1061-1946

**Abstract:** Tree stem detection is a key step toward retrieving detailed stem attributes from terrestrial laser scanning (TLS) data. Various point-based methods have been proposed for the stem point extraction at both individual tree and plot levels. The main limitation of the point-based methods is their high computing demand when dealing with plot-level TLS data. Although segment-based methods can reduce the computational burden and uncertainties of point cloud classification, its application is largely limited to urban scenes due to the complexity of the algorithm, as well as the conditions of natural forests. Here we propose a novel and simple segment-based method for efficient stem detection at the plot level, which is based on the curvature feature of the points and connected component segmentation. We tested our method using a public TLS dataset with six forest plots that were collected for the international TLS benchmarking project in Evo, Finland. Results showed that the mean accuracies of the stem point extraction were comparable to the state-of-art methods (>95%). The accuracies of the stem mappings were also comparable to the methods tested in the international TLS benchmarking project. Additionally, our method was applicable to a wide range of stem forms. In short, the proposed method is accurate and simple; it is a sensible solution for the stem detection of standing trees using TLS data.

**Keywords:** tree stem extraction; terrestrial laser scanning; segment-based classification; connected component segmentation

## 1. Introduction

Forests are important ecological and economic resources because they contain 80% of the earth's plant biomass [1], which is important to understand the global carbon balance [2]. The stem generally accounts for more than 65% of the tree biomass [3] and is thereby considered crucial for the accurate determination of tree-level aboveground biomass. In forestry, retrieving detailed stem attributes, such as stem curve, is essential for determining the inflection points or cut points along the stem, calculating the total and merchantable stem volume, and evaluating the quality of stems [4]. The detailed stem attributes can also improve the accuracy of biomass estimation of individual trees [5,6]. Traditionally, the stems are manually measured on-site using tapeline or caliper, which is time-consuming and

labor-intensive. Consequently, many different stem curve models have been developed for various tree species [7]. However, these models cannot be used to describe irregular stems.

Terrestrial laser scanning (TLS) is a robust technology to capture detailed three dimensional (3D) point clouds. Its application to the measurement of detailed stem attributes has been well demonstrated [8]. For the reconstruction of the stems and retrieval of stem attributes from the point clouds, tree stem detection is a key step in data processing. In some studies, tree stem detection is defined as the process of identifying the location of stems in a forest plot [9]. The stem locations can be detected from a layer or multilayered sliced points of certain heights above ground through circle-fitting methods [10–12]. Wang et al. [13] proposed a two-layer projection method based on the planar density and median z-normal value for the detection of stem locations. Once the stem locations are obtained, individual trees can be separated [11]. Then the stem attributes can be retrieved using circle fitting or cylinder fitting algorithms [14,15]. However, these stem detection methods often encounter specific difficulties in natural stands with a high density of stem and understory vegetation [16], and the retrieval of stem attributes can be influenced by the foliage points in leaf-on condition [11,17]. To deal with these challenges, some researchers have devoted their attention to directly identifying the stem or wood points from TLS data.

The intensity of laser returns, in theory, can be used to differentiate wood and leaf points as the intensity of leaf points is usually darker than wood [18]. However, in practice, the intensity of laser returns is not only related to the spectral property of targets but also the incidence angle, traveling distance [19,20], and surface roughness of the targets [21]. The calibration of the intensity value is complicated and time-consuming [22]. Danson et al. [23] tested a dual-wavelength full waveform TLS, and Li et al. [24] explored the feasibility of multi-wavelength LiDAR for the wood-leaf separation. However, there is no quantitative accuracy report on the point classification in their studies. The use of dual- or multi-wavelength and full-waveform information of TLS for the wood-leaf separation is limited by the availability of equipment. By contrast, the geometric features based on the 3D coordinates of points are more commonly used for the stem or wood points extraction.

Kelbe et al. [25] proposed an approach to detect stem points from the low-resolution single-scan TLS datasets based on planar projection and the planar point density distribution. The classification accuracies (i.e., the percentage of correctly detected stem points) reported in this study varied from 8% to 83% depending on the plot condition. Liang et al. [26] proposed a method to identify stem points using the local geometrical features, i.e., the flatness and normal direction. A point was recognized as a stem point if it has a low variance in one direction in the local coordinate system and has a nearly horizontal normal vector in the real-world coordinate system. Lalonde et al. [27] introduced a method that characterizes a point cloud using three salient features, i.e., scatter, linear, and surface. They characterized the features of the target objects by fitting a Gauss mixed model on manually labeled training data and used a Bayesian classification method to label the entire point cloud. Ma et al. [28] improved Lalonde's method by adding two additional filters based on geometric information, and the overall classification accuracy was improved from 84.28% to 97.8%. Moreover, Brodu and Lague [29] applied the local geometric features to multiple scales. The classifiers, including least absolute deviation and support vector machine (SVM), were used in supervised classification. Similarly, Xia et al. [30] adopted two-scale pointwise geometric features to identify the candidate stem points. Zhu et al. [31] combined various local geometric features and radiometric features to separate foliar and woody materials by using a random forest algorithm. The reported overall classification accuracy in their study ranged from 80% to 90%. Chen et al. [32] also applied the SVM classifier along with various features, such as local geometric features, intensity value, and normal vectors.

The previous methods based on the local geometric features have achieved satisfactory classification accuracies, although there is still room for improvement. A common classification strategy of the previous methods is to identify the stems from TLS data point by point. Compared with the segment-wise classification, these point-based methods require more computing resources [9]. These computational burdens are mainly due to two reasons. First, the geometric features need to

be calculated at each point by its surrounding points within a neighborhood. Second, the geometric features are varying with the radius of the neighborhood. Thereforee the latest algorithms usually extract two-scale (sometimes even multi-scale) geometric features [29,30], which greatly increase the computational cost.

The segment-based classification method can potentially reduce the computational burden and uncertainties of point cloud classification [33]. Compared with the point-based method, the segment-based method first divides a point cloud into segments, then classifies the segments based on the features of each segment, such as the statistical features of the points in a segment, and the size or shape of a segment [34]. A point cloud can be segmented using different methods [35], such as shape detection [36], region growing [37], and connected component segmentation [35]. In an urban environment, the segment-based method has been widely used [34]. In a forest environment, Polewski et al. [38] proposed an approach that combined a point-based and a segment-based method to detect fallen tree stems from TLS datasets. Amiri et al. [39] applied Polewski 's method to detect tree stems from very-high density airborne laser scanning datasets. However, they employed a supervised method in the stage of point-based classification by using multiple features, which increased the complexity of the algorithm and turned out to be a time-consuming approach. Olofsson et al. [40] recognized stem points by calculating the flatness feature of a subset of a point cloud within a voxel cell instead point by point, which has a much higher computational speed, but they voxelized the point cloud rather than segmented the point cloud.

In summary, the segment-based method for point cloud classification was widely used in urban scenes. However, its application is largely limited to urban scenes due to the complexity of the algorithm as well as the conditions of natural forests. In this study, we aim (1) to develop a new segment-based method for stem detection using a connected component segmentation algorithm, (2) to explore the method of point thinning before connected component segmentation based on the curvature feature points, (3) to evaluate the accuracy of stem point extraction using the proposed method, and (4) to evaluate the accuracy of stem mapping based on the extracted stem points from the proposed method.

## 2. Materials and Methods

### 2.1. Terrestrial Laser Scanning Data

The TLS datasets used in this study were obtained from seven sample forest plots, of which six sample plots were public datasets and provided by the international TLS benchmarking project [41]. The TLS benchmarking project was conducted by the Finish Geospatial Research Institute to comprehensively evaluate the capacity of the TLS data for forest inventory and the algorithm performances of the data processing for tree attribute extraction. Eighteen partners from Asia, Europe, and North America participated in the project and submitted their retrieved tree attributes for evaluation. The TLS data of the TLS benchmarking project was collected from twenty-four sample plots with different tree species, developing stages, stem densities, and abundance of understory vegetation in southern boreal forests in Evo, Finland (61.19°N, 25.11°E), and six of them with field reference data are open for nonprofit research purposes. These sample plots were classified into three complexity categories, i.e., easy, medium, and difficult, according to the density of stems and understory vegetation, which also reflect the level of complexity in TLS data processing. The category "easy" represented sample plots that had clear stem visibility with minimal understory vegetation and low stem density, "medium" represented sample plots with moderate stem densities and sparse understory vegetation, and "difficult" represented sample plots that had high stem densities and dense understory vegetation. These sample plots were all located in flat areas, each of them had a fixed size, 32 × 32 m, and was scanned both in single- and multi-scan mode using a Leica HDS1600 (Leica Geosystems AG, Heerbrugg, Switzerland) terrestrial laser scanner. The multi-scan TLS datasets consist of five scans located at the center and in four quadrantal directions of each sample plot, and

the single-scan datasets were just the center scan of multi-scan datasets. The dominant tree species are Scots pine (*Pinus sylvestris* L.), Norway spruce (*Picea abies* L. Karst.), silver birch (*Betula pendula* Roth), and downy birch (*Betula pubescens* Ehrh.).

The trees in the six public sample plots usually had straight up stems. To verify the applicability of the proposed method to various stem forms, we introduced another 50 × 50 m sample plot located in a pure Chinese scholar tree (*Styphnolobium japonicum*) forest in Beijing, China (39°46′N, 116°13′E), hereafter referred to as plot 7. The Chinese scholar tree generally had irregular and diverse stem forms, as shown in Figure 1. The slope of this forest area was less than 5°. The stem density of plot 7 was about 366 stems/ha. The mean diameter at breast height (DBH) and tree height was 28.1 cm and 12.7 m, respectively. Also, plot 7 had sparse understory vegetation. TLS dataset of the plot 7 was acquired in the multi-scan mode using a Riegl VZ-1000 (Riegl GmbH, Horn, Austria) terrestrial laser scanner; five scans were implemented at the center and at the center and in four quadrantal directions. There were about 128 million points contained in the TLS dataset of plot 7. Table 1 displays the characteristics of the seven sample plots and TLS datasets.

**Table 1.** Characteristics of the sample plots and the TLS datasets.

| Plot ID | Complexity Categories | Stem Density (stems/ha) | DBH (cm) | Tree Height (m) | Point Population (million) | |
|---|---|---|---|---|---|---|
| | | | | | Single-Scan | Multi-Scan |
| 1 | Easy | 498 | 22.8 ± 6.6 | 18.7 ± 3.9 | 23.6 | 111 |
| 2 | Easy | 820 | 16.0 ± 6.9 | 13.7 ± 4.0 | 23.6 | 114 |
| 3 | Medium | 1445 | 14.8 ± 7.4 | 15.5 ± 6.8 | 23.7 | 120 |
| 4 | Medium | 762 | 19.6 ± 14.1 | 16.1 ± 10.2 | 27.4 | 129 |
| 5 | Difficult | 1279 | 14.3 ± 13.2 | 13.0 ± 7.0 | 23.7 | 125 |
| 6 | Difficult | 2304 | 12.3 ± 5.5 | 13.0 ± 6.3 | 22.7 | 111 |
| 7 | Easy | 366 | 28.1 ± 7.5 | 12.7 ± 3.4 | - | 128 |

**Note:** The complexity of forest plots was defined in the TLS benchmarking project according to the stem density and abundance of understory vegetation. The DBH refers to the diameter at breast height (1.3 m). The link to the first six datasets can be found at http://laserscanning.fi/tls-benchmarking-results/.

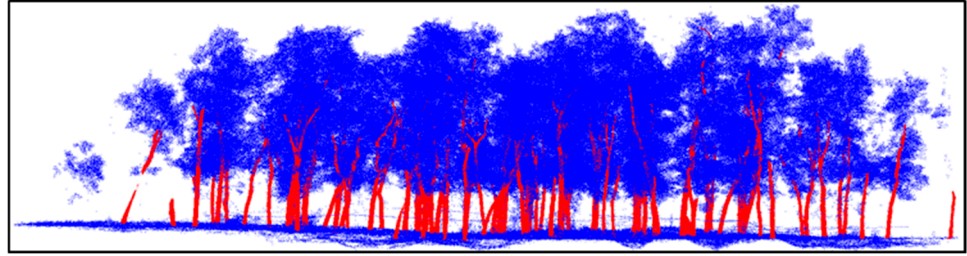

**Figure 1.** Illustration of the diverse stem forms of the plot 7. The stem points were displayed in red, and the non-stem points were displayed in blue.

*2.2. The Reference Data for Validation*

Along with the TLS data of the six sample plots, the TLS benchmarking project also provided the reference data of stem locations, diameter at breast height (1.3 m, DBH), and tree heights, which was used in this study to validate the accuracy of stem detection at the attribute level. The referential tree stem locations were gathered by integrating manual measurements from the multi-scan datasets and in the field. First, a preliminary stem location map was manually measured from multi-scan TLS data for trees having a complete point cloud. Then the preliminary stem location map was verified in situ, and the location of omitted stems was determined via the distances and directions of the omitted stem to its four neighboring known stems. The DBH was measured for each tree using steel calipers in the field. Tree height was measured with Vertex 3.0 (Haglöfs, Långsele, Sweden) to a resolution of 0.1 m. The expected accuracy of tree-height measurement was 0.5 m. For more details about the method of collecting referential tree level attributes, please refer to Liang et al. [41].

In order to validate the accuracy of stem point extraction at the point level, we used the manually extracted stem points as reference. The reference point clouds were classified into two parts, i.e., the stem points and non-stem points. It should be noted that the ground points in the reference TLS datasets have been filtered using an automated method, the cloth simulation filtering (CSF) proposed by Zhang et al. [42]. The details for the implementation of the CSF will be described in the following section. In the referential TLS data, the stems of understory trees that the DBHs were not greater than 5 cm and shrubs were considered as non-stem points to match the demand for forest inventory.

## 2.3. Tree Stem Point Extraction

The principle behind our method is the different spatial distribution characteristics between branches and foliage (B&F) points and stem points. In theory, the B&F points are spread over the entire canopy layer of a plot, which usually has small clusters due to their small point density and gaps within the canopy. By contrast, stem points generally have large clusters and vertical shapes.

The method for the stem point extraction had the following steps: (1) removal of the ground points, (2) B&F points thinning to enlarge the difference of point density between the stem points and B&F points, (3) connected component (CC) segmentation and stem segment identification, and (4) refinement of the extracted stems for removal of the remaining twigs. Figure 2 shows the overview of processing pipeline of the proposed method for the detection of tree stem points. The following sections describe the principle and detailed methods of the preceding steps mentioned.

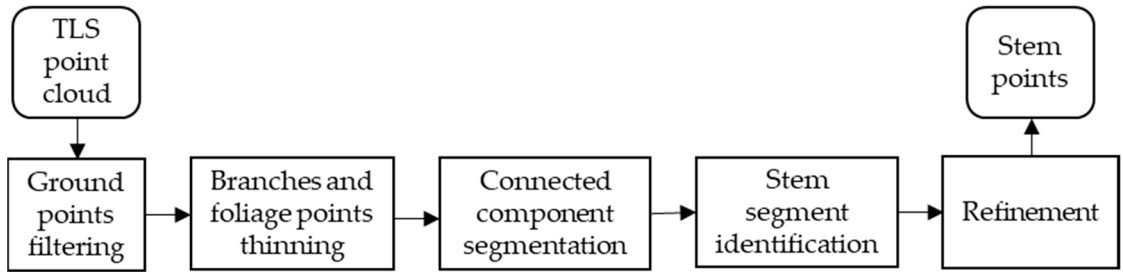

**Figure 2.** Overview of the proposed segment-based method for detecting tree stem points.

### 2.3.1. Ground Filtering

In our method, ground filtering was a prerequisite step for reducing the data volume and breaking the ground connection between stems. For fast processing and reliable results, we adopted the cloth simulation filtering (CSF) method in this study.

The CSF was a ground filtering algorithm based on the principle of cloth simulation [42]. By iteratively dropping a simulated cloth to an inverted (upside-down) LiDAR point cloud, the simulated cloth sticks to the ground points and bridge over the object points due to a certain degree of cloth hardness. The simulated cloth was modeled as a set of regular particles with mass. The interconnection between particles were "virtual springs". In each iteration, the particles dropped down under the force of gravity and then sprung back under the spring force. Finally, the ground points were detected through a pre-set distance threshold between points and the final settled cloth.

The CSF has been integrated into CloudCompare [43] as a third-party plug-in. Therefore, we implemented the CSF in CloudCompare in this study. First, the TLS point clouds were imported into CloudCompare, and the CSF was directly executed without any preprocessing steps. The cloth resolution and distance threshold (classification threshold) of the CSF were both set to 0.1 m, and the max iterations were set to 50 for all sample datasets. Then, the off-ground point clouds produced by the CSF were delivered to the next step as the inputs.

2.3.2. Branches and Foliage Points Thinning Based on Curvature Feature

The B&F points thinning procedure aims to reduce their point density through removing a part of B&F points and, in the meantime, retaining the stem points as much as possible. In our method, the normal change rate (NCR) was used as a curvature feature to identify the B&F points. The NCR, also known as the surface variation, is a measure of the surface curvature [44]. The calculation of the NCR was based on principal component analysis of the points within a local neighbourhood [45,46]. For a point $p_i = (x_i, y_i, z_i)$ in a point cloud $P = \{(x_i, y_i, z_i)\}_{i=1}^n$, the covariance matrix $C$ is defined as

$$C_{3\times3} = \frac{1}{n}\sum_{i=1}^{n}(p_i - \overline{p})(p_i - \overline{p})^T \tag{1}$$

where $\overline{p} = (1/n)\sum_{i=1}^{n}(p_{xi}, p_{yi}, p_{zi})$ is the centroid of P.

Then the eigenvalues $\lambda_k$ $(k = 2, 1, 0)$ of the covariance matrix $C$ were computed by using singular value decomposition (SVD) and sorted in descending order as $\lambda_2 \geq \lambda_1 \geq \lambda_0$. The eigenvalues were used to get the NCR at $p_i$:

$$NCR(p_i) = \frac{\lambda_0}{\lambda_0 + \lambda_1 + \lambda_2} \tag{2}$$

The curvature feature of a point cloud has already been used to separate tree components, e.g., stem, branches, and foliage in some studies [40,47]. The curvature on the stem points was usually smaller than that on B&F points (Figure 3) because stems are generally of large diameters (small curvature), and the point distribution on the stems was more regular. By contrast, branches had small diameters (large curvature) and are surrounded by foliage, and point distribution on foliage was statistically scattered relative to stems and branches. However, the curvature feature cannot utterly differentiate stem and B&F points in practice because of the complicated distribution of foliage points, some of which may also have small surface curvatures (Figure 4). Therefore, our purpose in this step was to remove a part of B&F points, instead of all of them, which could be reached by using a suitable NCR threshold. The points exceeding the NCR threshold were labeled as the B&F points and removed.

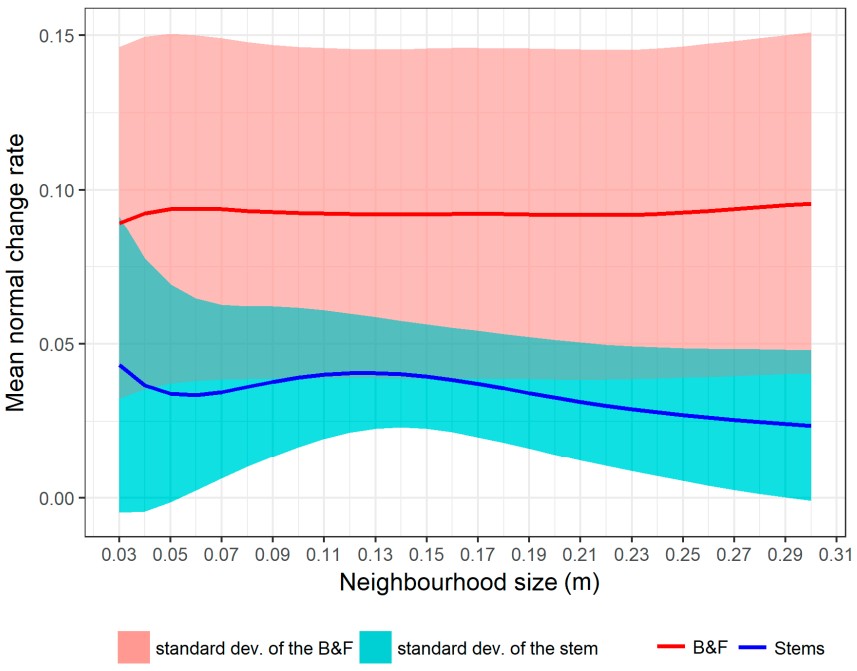

**Figure 3.** Mean normal change rate (NCR) of stems and branches and foliage (B&F) points from the multi-scan point cloud of the plot 1 with the increasing neighborhood size. The mean NCR on the stems was smaller than B&F. However, they cannot be completely separated by a threshold because of the deviation of the NCR on them.

For calculating the NCR of a point, the points near it within a neighbourhood were first found. Then, the NCR of the target point was calculated using Formula (2) based on its neighbourhood points. Finally, the NCR of all input points was obtained after traversing every point. In this study, the input point clouds for NCR calculation were the off-ground point clouds from the ground filtering step. We implemented this step using the batch script written in MATLAB (R2017b, The MathWorks, Ins.) with the curvature calculation function provided by CloudCompare (2.10.beta, 2018). Two parameters needed to be decided in the B&F points thinning procedure, i.e., the neighbourhood size for NCR calculation and the NCR threshold for labelling B&F points. The neighbourhood size was set as 5 cm, and the NCR threshold was set as 0.1. The 5 cm was the lower limit of the DBH that a tree was counted as having in this study. Approximately 5 cm was considered suitable in some previous studies for calculating local geometric features for the stem separation in forest plots [30,31,48]. The value of the NCR threshold was determined to retain the stem points but remove the B&F points as much as possible. An NCR threshold value of 0.1 can potentially retain almost all stem points and remove approximately half of the B&F points according to our tests. In practice, the NCR threshold can also be easily and quickly tuned via visual inspection before automated processing. The setting of the neighbourhood size was related to the minimum DBH of the stems in a plot rather than the point density and stand conditions. The NCR of points had a fixed range from 0 to 1/3 for any TLS point cloud, and the NCR threshold was a relative value rather than absolute value for different datasets. Therefore, the same set of parameters was adopted for all TLS datasets.

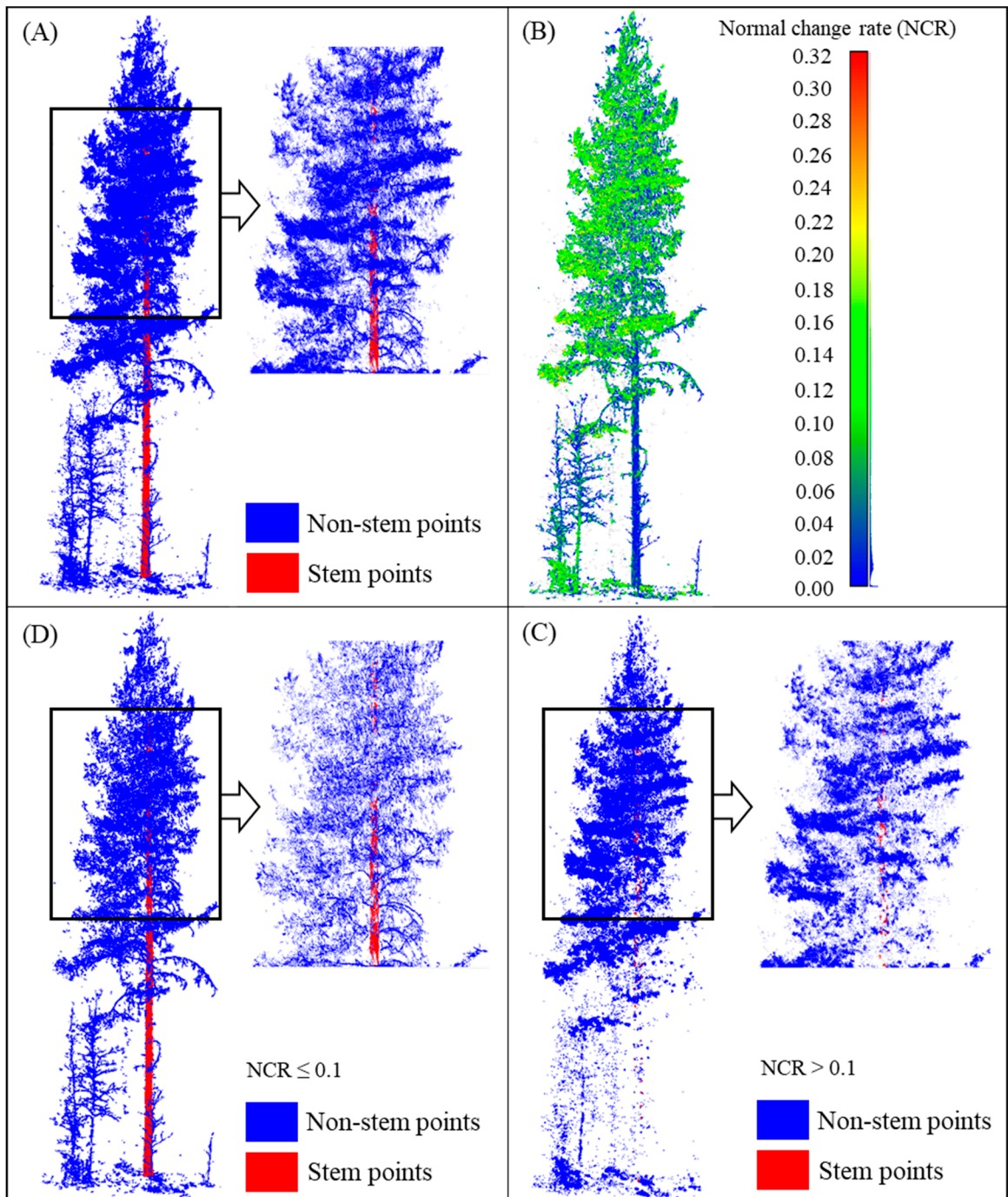

**Figure 4.** Illustration of the branches and foliage (B&F) points thinning based on normal change rate (NCR). (**A**) The original point cloud. (**B**) The NCR of each point was calculated using a 5 cm neighborhood size. (**C**) The points with an NCR exceeding 0.1 were removed from the original point cloud. (**D**) The remaining points where the NCR did not exceed 0.1. The figure shows that the original point cloud was thinned after removing a part of B&F points based on a threshold of NCR. Most stem points were retained, but a few of them were wrongly removed.

### 2.3.3. Segment-Based Stem Point Identification

Tree stem point extraction based on the connected component segmentation is the core step of the proposed method. The point cloud after B&F points thinning, as the input data, was processed as follows: (1) connected component segmentation; (2) calculating geometric features of each segment, i.e., the number of points (NoP) within the segment and the height-to-width (H-W) ratio of the segment;

and (3) stem segment identification based on the threshold of NoP and H-W ratio. Finally, all stem segments were merged into a single point cloud. Figure 5 shows the overview of the processing pipeline of the segment-based stem point identification.

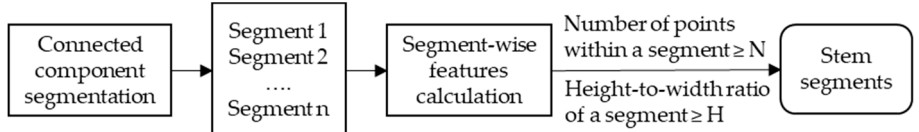

**Figure 5.** Overview of the step of segment-based stem point identification. N refers to the threshold of the number of points, and H refers to the threshold of height-to-width ratio.

We implemented this step using a batch script written in MATLAB (R2017b, The MathWorks, Ins.) with the connected component segmentation function provided by CloudCompare (2.10.alpha). In particular, we modified the source code of connected component segmentation function of CloudCompare to make the calculation of the H-W ratio possible. The following subsections present the details of this step.

Connected Component Segmentation

The CC segmentation based on the proximity of points is a simple and fast algorithm for the point cloud segmentation. First, the point cloud was voxelized using 3D grids. The grids that contained at least one point were assigned 1. Otherwise, the vacant grids were assigned 0. Then, the points in the adjacent grids with a value of 1 would be merged into the same segment. The vacant grids with a value of 0 became the gaps between the segments [49]. The grid size of the 3D grids was also the minimum distance between two segments.

The CC segmentation could not be directly applied to the TLS data of forest stands for the classification purpose because of the high connectivity between the stem and B&F points. Therefore, the step of B&F points thinning was regarded as a prerequisite that makes the CC segmentation effective for the stem point extraction. After the B&F points thinning, the point clouds were more likely to be separated into segments that contained points belong to a single class, as shown in Figure 6. In addition, the grid size of the CC segmentation should be small enough for the generation of segments. However, too small of a grid size may lead to producing too many small segments that would impact not only the time efficiency but also the classification accuracy of the proposed method. As a tradeoff, the grid size was set as 1 cm and applied for all datasets in this study.

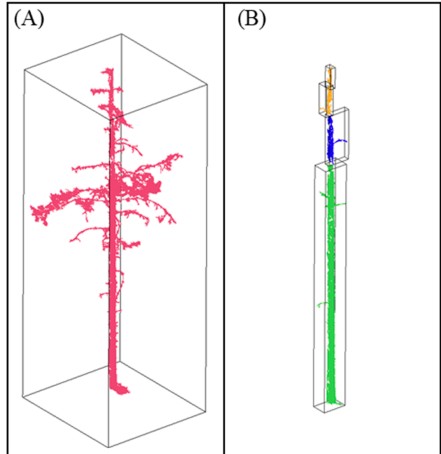

**Figure 6.** An example of two sets of stem segments produced by connected component segmentation using the same grid size of 1 cm. (**A**) The stem segments produced using the original point cloud. (**B**) The stem segments produced by the point cloud after branches and foliage points thinning. The colors were assigned to each segment randomly. The black boxes were the boundary of each segment. The figure shows that the stem segments of the point cloud after branches and foliage points thinning were separated from branches and foliage, which made the segment-based classification possible.

Segment Filtering by the Number of Points Per Segment

Tree stems generally return high density and large point clusters due to their large diameters, solid structure, and regular shape. Branches and foliage usually return small point clusters due to their small size, scattered distribution, canopy gaps, and occlusions. Therefore, small segments were more likely to be the B&F segments, especially when the B&F points had already thinned in the previous step (see Section 2.3.2).

Along with the CC segmentation, small segments in which the NoP was less than a threshold were directly removed. The threshold, as an input parameter of the CC segmentation algorithm, defined the minimum NoP per segment. The output segments from the CC segmentation were already filtered by the NoP threshold. The NoP threshold of the segments was decided according to the point density. For the multi-scan datasets, stem segments usually contain hundreds of thousands of points; we set 1000 as the NoP threshold. For the single-scan datasets, stem segments usually contain tens of thousands point, so the NoP threshold was set as 100 for them. The NoP thresholds were set as ten times smaller than the NoP of regular stem segments because the occlusions in forest stand may lead to fragmentation of the stem segments. A larger NoP threshold can filter out more B&F segments but also some small stem segments.

Stem Segment Extraction Based on the Height-To-Width Ratio

Although a large portion of the B&F segments were removed by using the NoP threshold, it was insufficient to remove all B&F points because some artificial features, thick branches, and plants that were close to the scanning center possibly had a large number of points in a segment, which cannot be removed by only using the NoP threshold. Fortunately, the stem segments on the standing trees had a prominent characteristic on the shape, which was a vertical and slim rectangle (Figure 6). The H-W ratio of the stem segments was larger than that of the B&F segments. Therefore, the H-W ratio was used to label stem segments. In our method, the H-W ratio was calculated using the standard deviation of 3D coordinates (stdev.X, stdev.Y, and stdev.Z) of all points in a segment. Z was the direction of height. The standard deviation was used to reduce the impact of outlier points. We calculated the H-W ratio as follows:

$$\text{H} - \text{W ratio} = \frac{stdev.Z}{\sqrt{(stdev.X)^2 + (stdev.Y)^2}} \tag{3}$$

where H-W ratio denoted the height-to-width ratio; and stdev.X, stdev.Y, and stdev.Z indicated the standard deviation of 3D coordinates of all points in a segment.

As shown in Figure 5, the H-W ratio was calculated for each segment after the CC segmentation, and the H-W ratio threshold was used to identify the stem segment. We integrated the calculation of H-W ratio and segment filtering into the CC segmentation function of CloudCompare (2.10.alpha). Therefore, the output segments from the CC segmentation were already filtered by the H-W ratio threshold. In this study, the same H-W ratio threshold (1.5) was set for all TLS datasets. A flowchart of the stem point extraction based on CC segmentation is shown in Figure 7.

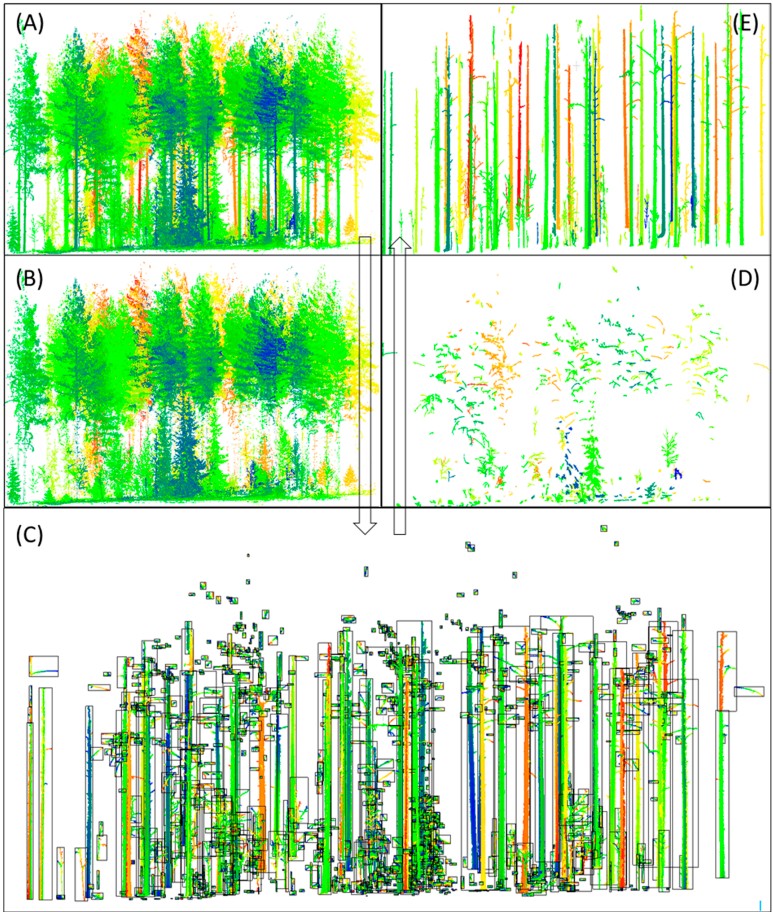

**Figure 7.** Flowchart of the stem point extraction based on connected component (CC) segmentation, taking the multi-scan dataset of the plot 1 as an example. (**A**) The preprocessed point cloud after ground filtering, and branch and foliage thinning. The following CC segmentation was implemented based on it. (**B**) The removed small segments that the number of points (NoP) per segment less than the threshold. (**C**) The high-probability stem segments with a NoP per segment larger than the threshold. (**D**) The removed segments with a height-to-width ratio less than the threshold. (**E**) The finally detected stem points by the CC segmentation method. In (**A**), (**B**), (**D**), and (**E**), the colors were randomly assigned to the points according to their X coordinate. In (**C**), the colors were randomly assigned to each segment, and the black boxes were the boundaries of each segment.

### 2.3.4. Refinement of Extracted Stem Points

Removal of the remaining branches and twigs in the extracted stem points is helpful for the further attribute retrieval of the stems. A stem point refining method based on point counting was applied in this step.

First, the detected stem segments were merged into a point cloud. Then the point cloud was vertically projected and rasterized with a pre-set grid step size (0.03 meters in this study). After that,

the points belonging under each grid cell were automatically counted. Given that the points on stems were usually vertically distributed, the grid cells over them always included a large number of points. By contrast, the grid cells over branches and foliage always included much fewer points than stems because most points on branches and foliage were removed in previous steps, and they were usually horizontally distributed. Therefore, the grid cells that the number of points less than the threshold were labeled as branches and foliage cells, and the points belonging in these cells were removed (Figure 8). The threshold of the number of points in this step was set based on the mean value of the Gauss distribution of the number of points pre-cell. After the stem-refining step, most remaining branches and twigs could be removed (Figure 9).

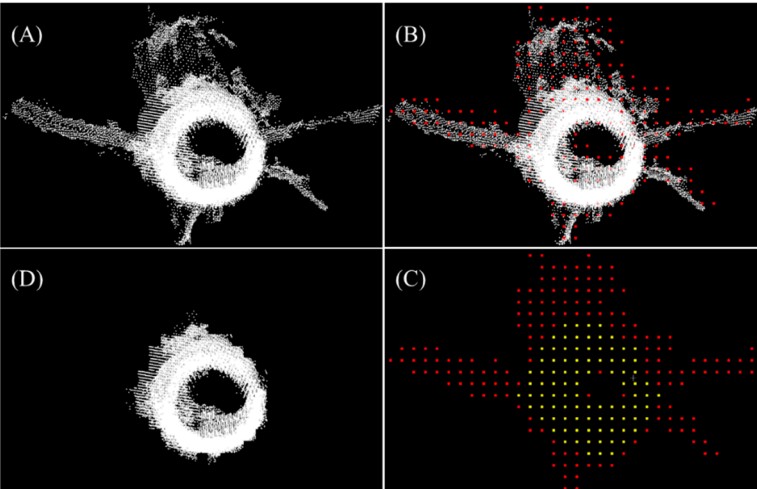

**Figure 8.** Illustration of the refinement step. (**A**) The points of an extracted stem were vertically projected. (**B**) Then, the projected points were rasterized. The red points indicate the northeast corner of the grid cells. (**C**) The points belonging to each grid cell were automatically counted. The grid cells with an NoP larger than the threshold were labeled as stem cells, which were indicated by the yellow points here. (**D**) The points remaining after the refinement.

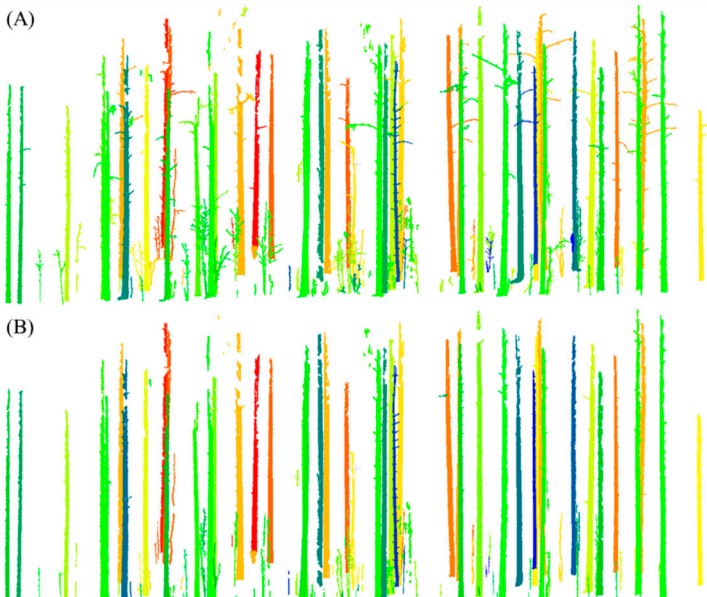

**Figure 9.** Illustration of the result after CC-stem-extraction (**A**) and the result after refinement (**B**).

### 2.4. Segmentation of Individual Stems and Stem Mapping

To obtain the individual stems, we used the CC segmentation algorithm again on the point clouds of the detected stems from previous steps. However, in this step, the grid size for the CC segmentation was set based on the distance between the two closest stems. Therefore, a larger grid size (0.1 m) than the previous step was adopted for all TLS datasets. It was easy to segment the stem point cloud into individual stems since the vast majority of the non-stem points were removed.

For each stem, the stem diameters were measured at the different target heights above the ground, starting at 0.65 m and then by 1.3 m, 2 m, and then for every next meter, until the maximum measurable height was reached [26]. A randomized Hough transform (RHT) for the circle detection [50] was applied to measure the diameters at different heights along the stem. We assumed that the stems were approximately cylindrical in this study, which is a very common assumption in forest inventory tasks [4], although the stem form was not strictly circular but an ellipsoid in some cases. The RHT randomly selects three points from a point slice and calculates the circle parameter. This process is performed iteratively with a fixed number of iterations (200 in this study). An accumulator is used to record these circle parameters. If the circle is similar to a circle in the accumulator, we replace the existing circle with the average of both circles and add 1 to its score. Otherwise, we insert the circle into an empty position in the accumulator and assign a score of 1. Finally, the circle with the highest score is selected and the horizontal coordinates of the fitted circles' center were averaged to be the position coordinates of the stem. The diameter at 1.3 m was the DBH; however, if the DBH could not be obtained due to data missing, the diameters at other heights would be averaged and regarded as the DBH. The stems with a DBH larger than 5 cm were classified as detected stems.

### 2.5. Methods of Accuracy Evaluation

The accuracy of stem-point extraction was assessed by counting the percentage of error-classifying points against the reference TLS datasets. Three types of error were quantified in this study: type I error, type II error, and total error. The type I error (T.I) was defined and calculated using the number of stem points that were wrongly removed divided by the number of referential stem points, which is also known as the omission errors. The type II error (T.II) was the number of B&F points that were wrongly classified as stem points divided by the number of referential B&F points, which is also known as the commission errors. The total error (T.E.) was the number of wrongly classified points divided by the total points. Finally, the total accuracy (T.A.) was described by the percentage of correctly classified points, which is complementary with the total error.

In accordance with the TLS benchmarking project [41], the accuracy of stem detection was assessed using the completeness, the correctness, and the mean accuracy of detection; The completeness measures the percentage of the reference trees is found using the proposed method. The correctness measures the percentage of the trees extracted using the proposed method is correct. The mean accuracy is the joint probability that a detected stem randomly chosen detection is correct detection. They are defined as:

$$\text{Completeness} = \frac{n_{\text{match}}}{n_{\text{ref}}} \tag{4}$$

$$\text{Correctness} = \frac{n_{\text{match}}}{n_{\text{extr}}} \tag{5}$$

$$\text{Mean accuracy of detection} = \frac{2n_{\text{match}}}{(n_{ref} + n_{extr})} \tag{6}$$

where $n_{\text{match}}$ is the number of found reference trees, $n_{ref}$ is the number of reference trees, and $n_{extr}$ is the number of trees detected.

All tests were conducted on a typical desktop computer with an Intel Core i7-6700 CPU (3.40 GHz) and 16 GB RAM.

## 3. Results

### 3.1. The Accuracy of Stem Point Extraction

Table 2 shows the accuracy of the stem point extraction, i.e., the type I error (T.I), type II error (T.II), total error (T.E.), and total accuracy (T.A.), which have been defined in Section 2.5. The results showed high accuracies of the stem point extraction across all sample plots both for single-scan and multi-scan TLS datasets. The mean total accuracy of the multi-scan datasets was 96.29%, and the accuracies of different plots were very consistent as the standard deviation of the total accuracies was only 0.53%. The mean total accuracy of the single-scan datasets was 95.81%, which was slightly lower than that of multi-scan datasets, so does the consistency of the total accuracy between different plots (2.16%). Since the sample plots had different complexities and stem forms, the consistency of the total accuracy between them indicates that the proposed method was adapting well to the different forest conditions and stem forms. It is worth mentioning that plot 7 obtained the best total accuracy, although it contained slanted and curving stem forms. Figure 10 shows the stem segments of plot 7 recognized by the H-W ratio threshold, which were correctly reserved.

**Table 2.** The accuracy of stem point extraction using the proposed method.

| Plot ID | Multi-Scan | | | | Single-Scan | | | |
|---|---|---|---|---|---|---|---|---|
| | T.I(%) | T.II(%) | T.E.(%) | T.A.(%) | T.I(%) | T.II(%) | T.E.(%) | T.A.(%) |
| 1 | 5.54 | 2.87 | 3.46 | 96.54 | 4.86 | 4.62 | 4.77 | 95.23 |
| 2 | 7.97 | 3.03 | 3.96 | 96.04 | 10.68 | 0.79 | 2.43 | 97.57 |
| 3 | 2.28 | 5.04 | 4.18 | 95.82 | 2.82 | 8.44 | 6.68 | 93.32 |
| 4 | 2.19 | 4.87 | 3.85 | 96.15 | 2.45 | 5.91 | 4.17 | 95.83 |
| 5 | 10.64 | 2.08 | 3.19 | 96.81 | 3.44 | 0.38 | 1.03 | 98.97 |
| 6 | 12.83 | 2.96 | 4.38 | 95.62 | 10.72 | 4.96 | 6.08 | 93.92 |
| 7 | 6.49 | 2.42 | 2.92 | 97.08 | - | - | - | - |
| Mean | 6.85 | 3.32 | 3.71 | 96.29 | 5.83 | 4.18 | 4.19 | 95.81 |
| Std.Dev | 4.00 | 1.16 | 0.53 | 0.53 | 3.87 | 3.10 | 2.16 | 2.16 |

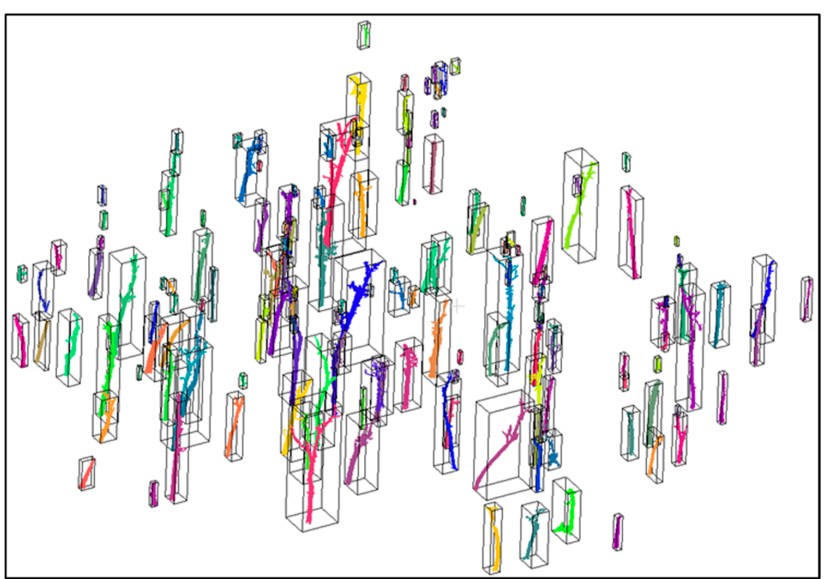

**Figure 10.** Tree stem point extraction of the plot with diverse stem forms using the proposed method. The figure shows the remained segments after connected-component segmentation and segment filtering using a height-to-width ratio threshold (1.5). The segments were assigned random colors. It demonstrates that the segments of slanted and curving stems were correctly reserved, which means that the height-to-width ratio of all stem segments was larger than the specified height-to-width ratio in this test.

### 3.2. The Accuracy of Stem Mapping

Table 3 shows the accuracy of stem mapping of the sample plots using both multi- and single-scan TLS datasets. The results demonstrated that the accuracy of stem mapping using multi-scan TLS datasets was distinctly superior to using single-scan TLS datasets from all accuracy indicators. The complexity of the forest stand, i.e., the stem density and abundance of understory vegetation, severely affected the accuracy of stem mapping regardless of the scan mode of TLS data collection. The completeness of stem detection consistently decreased with the increasing of forest stand complexity, and so did the mean accuracy of stem detection. However, the correctness of stem detection remained high and not relevant to the scan mode and forest stand complexity. This was mainly because the non-stem points had been accurately removed, and the correctness of stem detection was not likely influenced by the non-stem components in the forest plots. The correctness was not 100% in some plots because a few of the stems of understory vegetation were wrongly recognized as tree stems. The results also showed accurate estimates of stem locations and DBHs. The mean bias of stem location was 2.22 cm and 4.12 cm for the multi- and single-scan datasets, respectively. The DBH was underestimated in all sample plots using both scan modes. The mean underestimation was 1.42 cm and 2.85 cm for the multi- and single-scan datasets, respectively.

**Table 3.** The accuracy of stem mapping.

| Scan Mode | Plot ID | Complexity Categories | Stem Detection | | | Location | | DBH | |
|---|---|---|---|---|---|---|---|---|---|
| | | | Completeness (%) | Correctness (%) | Mean Accuracy (%) | RMSE (cm) | Bias (cm) | RMSE (cm) | Bias (cm) |
| Multi-Scan | 1 | Easy | 86.27 | 97.78 | 91.67 | 0.90 | 0.73 | 1.60 | −0.57 |
| | 2 | Easy | 82.14 | 95.83 | 88.46 | 2.40 | 1.20 | 2.16 | −1.11 |
| | 3 | Medium | 61.49 | 100.00 | 76.15 | 3.57 | 2.64 | 2.98 | −1.8 |
| | 4 | Medium | 57.69 | 97.83 | 72.58 | 3.52 | 1.58 | 2.4 | −0.97 |
| | 5 | Difficult | 45.80 | 93.75 | 61.54 | 5.12 | 3.21 | 3.02 | −1.87 |
| | 6 | Difficult | 26.27 | 98.41 | 41.47 | 6.21 | 3.93 | 3.85 | −2.21 |
| | Mean | | 59.94 | 97.27 | 71.98 | 3.62 | 2.22 | 2.67 | −1.42 |
| Single-Scan | 1 | Easy | 80.39 | 97.62 | 88.17 | 7.10 | 5.10 | 4.20 | −1.96 |
| | 2 | Easy | 57.14 | 92.31 | 70.59 | 3.20 | 2.15 | 3.19 | −1.82 |
| | 3 | Medium | 46.62 | 100.00 | 63.59 | 6.86 | 5.45 | 4.43 | −3.67 |
| | 4 | Medium | 34.62 | 100.00 | 51.43 | 3.07 | 1.97 | 3.69 | −2.46 |
| | 5 | Difficult | 13.74 | 90.00 | 23.84 | 5.00 | 4.34 | 4.37 | −3.30 |
| | 6 | Difficult | 8.47 | 100.00 | 15.63 | 7.68 | 5.72 | 4.71 | −3.86 |
| | Mean | | 40.16 | 96.65 | 52.21 | 5.49 | 4.12 | 4.10 | −2.85 |

## 4. Discussion

### 4.1. Stem Detection Accuracy

This study proposed a novel and simple approach for the stem detection from TLS data, which achieved comparable accuracies (from 93% to 99%) of the stem point extraction compared with the reported accuracies in the literature, which ranged from 8% to 98% [25,31,48]. From the perspective of the stem mapping, the proposed method also achieved comparable accuracies to the eighteen methods, which have been tested in the TLS benchmarking project (Table 4). Although the stem points have been accurately extracted, the completeness of stem detection was very low in the difficult plots. This indicates that the complexity of the forest stand was the most important factor that influenced the accuracy of stem detection. For the accuracy of stem location, the maximum RMSE in each complexity and scan mode was similar among different methods (Table 5). In the TLS benchmarking project, the most robust methods delivered an RMSE of DBH ranging from 2 to 4 cm for all three complexity categories from the single-scan dataset and less than 2 cm from multi-scan datasets. In our method, the mean RMSE of DBH from single-scan and multi-scan were 4.10 cm and 2.67 cm, respectively.

**Table 4.** Comparison of the mean accuracy of stem detection between TLS benchmarking methods and our method.

| Complexity Categories | Benchmarking Methods | | Our Method | |
|---|---|---|---|---|
| | Single-Scan | Multi-Scan | Single-Scan | Multi-Scan |
| Easy | 75% | 80% | 79% | 90% |
| Medium | 64% | 74% | 58% | 74% |
| Difficult | 31% | 53% | 20% | 52% |

**Table 5.** Comparison of the maximum RMSE of stem location between TLS benchmarking methods and our method.

| Complexity Categories | Benchmarking Methods | | Our Method | |
|---|---|---|---|---|
| | Single-Scan | Multi-Scan | Single-Scan | Multi-Scan |
| Easy | 5 cm | 3 cm | 7.1 cm | 2.4 cm |
| Medium | 8 cm | 5 cm | 6.9 cm | 3.6 cm |
| Difficult | 10 cm | 9 cm | 7.7 cm | 6.2 cm |

### 4.2. Applicability to Various Stem Forms

The proposed method is also advantaged in its usability. The same set of parameters was applied to different plots with different complexities. However, the total accuracies of the stem point extraction were very consistent (Table 2). This means that the proposed method can be applied to the forest plots with various stand conditions, and the results were insensitive to input parameters. We performed the proposed method on an additional plot with diverse stem forms using a set of parameters. The results turned out to be very accurate (Table 2). This indicates that the slanted and curving stems did not impact on the stem segment labelling based on the H-W ratio threshold in this plot.

### 5. Conclusions

In this study, we proposed a novel and simple segment-based approach to detect tree stems from plot-level TLS data. The proposed method achieved comparable accuracies not only for the stem point extraction but also for the stem attribute estimation. Our method was applicable in various forest conditions with different species, stem densities, the abundance of understory vegetation, and stem forms. Our work verified the effectiveness of connected component segmentation for the classification of dense point cloud with the assistance of points thinning using the curvature feature in forest scenes. The method we proposed in this study is accurate and simple; it is a sensible solution for the stem detection of standing trees using TLS data. However, the variety of TLS data and forest plots were still limited in this study. It is unknown whether the proposed method can be applied to the sparse point cloud, which needs further studies.

**Author Contributions:** Conceptualization, P.W.; Data curation, S.C. and Y.C.; Funding acquisition, W.Z. and G.Y.; Methodology, P.W.; Project administration, G.Y.; Supervision, W.Z.; Writing—original draft, P.W.; Writing—review and editing, W.Z., T.W., and X.J.

**Funding:** This work was supported by the National Natural Science Foundation of China Grant Nos 41671414, 41331171 and 41171265. The National Key Research and Development Program of China (NO. 2016YFB0501404).

**Acknowledgments:** The authors would like to thank the editors and the anonymous reviewers for their valuable comments and suggestions, which have helped immensely in improving the quality of this paper.

**Conflicts of Interest:** The authors declare no conflict of interest.

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
