# Peer review of "A Novel Approach for the Detection of Standing Tree Stems from Plot-Level Terrestrial Laser Scanning Data"

_remotesensing, doi:10.3390/rs11020211_

Round 1
Reviewer 1 Report
You will find the comments in the attached file

Author Response
Please find the response to reviewer 1 comments in the attached file.

Reviewer 2 Report
Review of paper
A fast approach for the detection of standing tree stems from plot-level terrestrial laser scanning data
by
Wuming Zhang , Peng Wan, Tiejun Wang, Shangshu Cai, Yiming Chen , Xiuliang Jin, and
Guangjian Yan
The paper is in the scope of the journal and is written in a clear way (could benefit of some slight language editing).
The authors assume that many of the current algorithms use supervised learning and is in need of training data. This is simply not the case.
A main emphasis of the paper is that the algorithm is fast. But there is a lack of comparison with older techniques. How much faster is this method compared to the different TLS tree detection algorithms?
For instance page 2 row 89 claim that point based classification methods have a high computing demand and gives two references [24,25] that has no time evaluation at all.
The algorithm is new, however some similar techniques already exist in the literature. The authors should give credit to earlier work and put their work in relation to existing methods.
For instance
in the paper by:
Raumonen, P.; Kaasalainen, M.; Åkerblom, M.; Kaasalainen, S.; Kaartinen, H.; Vastaranta, M.; Holopainen, M.; Disney, M.; Lewis, P. Fast automatic precision tree models from terrestrial laser scanner data. Remote Sens. 2013, 5, 491–520.
the model is constructed by a local approach in which the point cloud is covered with small sets
corresponding to connected surface patches in the tree surface. The neighbor-relations and
geometrical properties of these cover sets are used to reconstruct the details of the tree and,
step by step, the whole tree. The segmentation part of this paper reminds of the authors version. This algorithm can complete a tree from 245-1160 seconds on a MacBook Pro, 2.8 GHz, 8 GB. The study is from 2013, PC:s are faster today.
In the paper by:
Olofsson and Holmgren. Single tree stem profile detection using terrestrial laser scanner data,
flatness saliency features and curvature properties. Forests. 2016, 7(9):1–23.
curvature properties and eigenvalues are used when segmenting the tree stems similar to the authors version. Also an angle criteria where used where flat surfaces perpendicular to the zenith direction are assumed to be part of a stem which will give the same effect as the H-W ratio used by the authors. In this paper, a point cloud saliency computation algorithm that increase the calculation speed with a factor of 100 is shown.
In the paper by:
Di Wang, Markus Hollaus, Eetu Puttonen and Norbert Pfeifer.Automatic and Self-Adaptive Stem Reconstruction in Landslide-Affected Forests Remote Sens. 2016, 8, 974
Point cloud density and median z-normal value is used when filtering the point cloud which will have an effect similar to the refinement step and the H-W ratio by the authors.
The paper by:
Liang, X.; Kankare, V.; Yu, X.; Hyyppä, J.; Holopainen, M. Automated Stem Curve Measurement Using Terrestrial Laser Scanning. IEEE Trans. Geosci. Remote Sens. 2014, 52, 1739–1748,
doi:10.1109/TGRS.2013.2253783.
Is already included by the authors but they should mention the similarities of their algorithm and this one.
The paper by:
Xia, S.; Wang, C.; Pan, F.; Xi, X.; Zeng, H.; Liu, H. Detecting stems in dense and homogeneous forest using single-scan TLS. Forests 2015, 6, 3923–3945.
is also included by the authors but they can explain the similarities to the authors version in a better way.
Author Response
Please find the response to reviewer 2 comments in the attached file.

Reviewer 3 Report
The authors developed a novel method to identify tree stems in forests and concluded that their method was more accurate and faster than others. The results showed the novel method had better accuracy. However, the authors did not design an appropriate experiment to compare their processing time to other studies. I would recommend the authors either apply their method and other study methods under the same environmental conditions and provide a statistical results or rewrite the manuscript and remove the statement of time efficiency of the stem point detection.
Test data section and methods section should combine methods and materials section.
The objectives of the study are not mentioned in the abstract and introduction. Please clarify the objectives of the study.
Line 73-74 the sentence should use past tense.
Line 139 "The main tree species..." should change to "The dominant tree species..."
Line 157 You mentioned you used an automated method to filter the ground points in the reference TLS dataset. Could you briefly describe the automated method and how did you apply?
4.1 Methods of accuracy evaluation-This section can move to the methods and materials section.
Line 350-356, 382-385 The paragraphs should move to the methods and materials section.
5.3 Applicability to various stem forms-This section is unexpected. You could combine this section in your study and present the results in the results section.
Table 3. The scan mode column needs to be consistent.
Table 5 and Table 6 have the same captions.
Author Response
Please find the response to reviewer 3 comments in the attached file.

Round 2
Reviewer 2 Report
I recommend the publication of this paper
Author Response
Thanks again for your review of our manuscript.
Reviewer 3 Report
I appreciate the authors' efforts to complete all of the corrections and improve the structure of the article. However, some English grammar and language use problems still need to improve to make the article understandable for readers. Besides, I still doubt the necessary of presenting the time efficiency of the stem point extraction. The configuration of a typical desktop computer can also affect your processing time of the stem point extraction, such as the RAM (DDR3 or DDR4), the hard drive (SSD or HDD), the graphics card (GPU) and etc. Therefore, you might need to think about it before you complete the final version of the article.
Author Response
Response: We appreciate the reviewer’s comments and suggestion. The manuscript has been further edited by a professional English editor. Regarding the presenting of the time efficiency issue, we agree that the configuration of a computer can severely affect the processing time. After careful consideration and much thought, we have decided to remove all “time efficiency” related paragraphs from our manuscript as suggested by the reviewer, including the Abstract, Introduction, Results, Discussion and Conclusion. We hope that the revised manuscript will meet the reviewer’s expectations. Thanks again for your review of our manuscript.